# Green Solutions to a Growing Problem: Harnessing Plants for Antibiotic Removal from the Environment

**DOI:** 10.3390/antibiotics14101031

**Published:** 2025-10-15

**Authors:** Gaia Cusumano, Giancarlo Angeles Flores, Roberto Venanzoni, Paola Angelini, Gokhan Zengin

**Affiliations:** 1Department of Chemistry, Biology and Biotechnology, University of Perugia, Via del Giochetto, 06122 Perugia, Italy; gaia.cusumano@dottorandi.unipg.it (G.C.); giancarlo.angelesflores@unipg.it (G.A.F.); roberto.venanzoni@unipg.it (R.V.); 2Centro di Ricerca per l’Innovazione, Digitalizzazione, Valorizzazione e Fruizione del Patrimonio Culturale e Ambientale (CE.D.I.PA.), Piazza San Gabriele dell’Addolorata, 4, 06049 Spoleto, Italy; 3Department of Biology, Science Faculty, Selcuk University, Konya 43130, Turkey; biyologzengin@gmail.com

**Keywords:** phytoremediation, antibiotic pollution, rhizosphere microbiome, constructed wetlands, antimicrobial resistance, plant–microbe interactions

## Abstract

Environmental dissemination of antibiotics is a pressing global challenge, driving ecological imbalances and the proliferation of antibiotic resistance genes (ARGs). Conventional treatment technologies often fail to fully eliminate these micropollutants or are cost-prohibitive for widespread use. In this context, phytoremediation—using plants and their associated microbiota to remove, transform, or immobilize contaminants—has emerged as an effective and promising, low-impact, and nature-based approach for mitigating antibiotic pollution in aquatic and terrestrial environments. This review provides a comprehensive synthesis of the physiological, biochemical, and ecological mechanisms by which plants interact with antibiotics, including phytoextraction, phytodegradation, rhizodegradation, and phytostabilization. This review prioritizes phytoremediation goals, with attention to high-performing aquatic (e.g., *Lemna minor*, *Eichhornia crassipes*, *Phragmites australis*) and terrestrial plants (e.g., *Brassica juncea*, *Zea mays*) and their ability to remediate major classes of antibiotics. This study highlights the role of rhizosphere microbes and engineered systems in phytoremediation, while noting challenges such as variable efficiency, phytotoxicity risks, limited knowledge of by-products, and environmental concerns with antibiotic degradation. Future perspectives include the integration of genetic engineering, microbiome optimization, and smart monitoring technologies to enhance system performance and scalability. Plant-based solutions thus represent a vital component of next-generation remediation strategies aimed at reducing antibiotic burdens in the environment and curbing the rise in antimicrobial resistance.

## 1. Introduction

Antibiotics have become indispensable in modern medicine and agriculture, playing a central role in treating infectious diseases and enhancing animal productivity. However, their widespread and often indiscriminate use has led to significant environmental contamination. Antibiotics enter natural ecosystems through various pathways—including wastewater discharge, agricultural runoff, landfill leachates, and pharmaceutical effluents—and are frequently detected in surface water, groundwater, sediments, and even agricultural soils [1,2].

Once in the environment, even at sub-therapeutic concentrations, antibiotics can disrupt microbial community dynamics, impair ecosystem functions, and, most critically, select for antibiotic-resistant bacteria and resistance genes [3,4]. Antimicrobial resistance (AMR) is now recognized as a global health crisis, contributing to nearly 5 million deaths in 2019 and projected to cause up to 10 million annual fatalities by 2050 if no action is taken [5].

Conventional wastewater treatment technologies—including activated sludge systems, membrane filtration, and advanced oxidation processes—are often ineffective in fully removing antibiotics, especially in rural or resource-constrained settings [6,7]. Moreover, these systems can be prohibitively expensive and energy-intensive and may produce secondary pollutants.

In this context, phytoremediation—the use of plants and their associated microbiota to extract, transform, or immobilize contaminants—has emerged as a sustainable and cost-effective alternative [8]. Plants are capable of antibiotic removal through direct uptake (phytoextraction), enzymatic degradation (phytodegradation), stimulation of rhizospheric microbial communities (rhizodegradation), and stabilization in the root zone (phytostabilization) [9,10,11].

Furthermore, engineered plant-based systems such as constructed wetlands and floating treatment wetlands have shown promising results in removing a wide range of antibiotics while also improving overall water quality and providing ecological co-benefits [12,13].

This review provides a critical synthesis of current knowledge on phytoremediation of antibiotics. It explores the physiological and biochemical mechanisms underpinning plant-based removal, highlights key terrestrial and aquatic plant species with remediation potential, and discusses the synergistic role of rhizospheric microbiomes. Additionally, we address current limitations and propose future directions, including biotechnological innovations, microbiome engineering, and integrated treatment systems. By advancing understanding in this area, phytoremediation may contribute significantly to nature-based solutions for mitigating environmental antibiotic pollution and curbing the spread of AMR.

## 2. Materials and Methods

### 2.1. Review Design and Objectives

This systematic review was conducted following the PRISMA (Preferred Reporting Items for Systematic Reviews and Meta-Analyses) guidelines [14], with the aim of synthesize current knowledge on the role of plants and plant-associated systems in the remediation of environmental antibiotic contamination. The objectives were to (i) identify relevant plant species and mechanisms involved in antibiotic removal, (ii) evaluate the performance and limitations of plant-based systems (including constructed wetlands), and (iii) highlight emerging strategies and research gaps in the field.

### 2.2. Search Strategy

A comprehensive literature search was performed across the electronic databases PubMed, Web of Science, and Scopus from 2015 through August 2025. The following Boolean string was used (adapted slightly per database): (antibiotic* OR “antimicrobial compounds”) AND (phytoremediation OR “plant-based remediation” OR “constructed wetlands” OR “plant–microbe interaction*” OR rhizodegradation) AND (environment* OR soil OR wastewater OR water OR “agricultural runoff”).

Additional references were retrieved through manual searches of the bibliographies of included articles.

### 2.3. Eligibility Criteria

Articles were selected based on predefined inclusion and exclusion criteria, as summarized in Table 1 below.

## 3. Why Plants? A Green Alternative for a Global Challenge

As the limitations of conventional wastewater treatment technologies become increasingly evident, the pursuit of low-cost, scalable, and sustainable alternatives has gained urgency. Advanced treatment processes—such as ozonation, UV photolysis, membrane filtration, and advanced oxidation processes—while effective, often require high energy input, produce toxic by-products, or rely on sophisticated infrastructure that is economically unfeasible for many developing regions and rural contexts [15,16]. Moreover, these technologies typically lack adaptability to handle the chemical diversity and low concentrations of antibiotics frequently encountered in wastewater effluents [17].

In this context, plant-based remediation strategies—collectively referred to as phytoremediation—are emerging as a viable and ecologically sound solution. Phytoremediation exploits the inherent physiological, biochemical, and ecological functions of plants to remove, transform, or immobilize environmental contaminants. Unlike conventional physicochemical approaches, phytoremediation systems operate in situ, avoid the use of external chemicals, and contribute positively to the local ecosystem by stabilizing soils, enhancing biodiversity, and reducing erosion [18].

Plants can address antibiotic contamination through several complementary mechanisms (Figure 1). Phytoextraction enables the uptake and accumulation of antibiotics within plant tissues [19], whereas phytodegradation involves enzymatic pathways—e.g., include hydroxylation of sulfonamides by peroxidases, demethylation of macrolides by cytochrome P450s, and oxidation of tetracyclines by laccases—that transform these compounds into less toxic metabolites [20,21]. Rhizodegradation occurs when root systems stimulate associated microbial communities capable of biodegrading antibiotics [11,22], while phytostabilization reduces mobility by immobilizing or adsorbing antibiotic molecules in the rhizosphere or on plant surfaces [23].

These processes often act synergistically within planted systems, particularly when supported by robust rhizosphere microbial communities and plant-secreted enzymes that accelerate antibiotic transformation [24,25]. Engineered phytoremediation approaches, notably constructed wetlands (CWs) and floating treatment wetlands (FTWs), have demonstrated substantial potential for large-scale implementation [26]. Such nature-based systems have already been successfully deployed to remove antibiotics including sulfamethoxazole, ciprofloxacin, and tetracyclines from contaminated water bodies [27,28]. Recent investigations emphasize the superior performance of hybrid systems that integrate phytoremediation with microbial or catalytic enhancements [10].

Despite these promising outcomes, plant-based remediation is not universally applicable. Its efficiency depends on several factors, such as plant species, antibiotic class and concentration, climatic conditions, and water chemistry [29]. Moreover, antibiotic uptake is strongly influenced by physicochemical properties including hydrophobicity, ionization potential, and molecular structure [30].

Nevertheless, when strategically combined with microbial symbionts or coupled with engineered treatment technologies—such as nanomaterials or solar photocatalysis—phytoremediation represents a compelling strategy to address the global challenge of antibiotic pollution. Its relative simplicity, cost-effectiveness, and environmental co-benefits establish plant-based systems as essential elements within the portfolio of sustainable wastewater treatment solutions [31].

In the following sections, we provide a critical examination of the mechanisms that underpin phytoremediation, assess high-performing plant species, and explore innovative applications together with the challenges posed by real-world implementation.

Phytoremediation encompasses a range of biologically mediated mechanisms through which plants interact with environmental pollutants, including pharmaceutical residues such as antibiotics [9,32]. Unlike conventional treatments that rely on physical separation or chemical degradation, phytoremediation exploits plant metabolism, root exudation, and plant–microbe interactions to transform or immobilize contaminants [33]. The efficiency of these processes depends on the physicochemical properties of the antibiotic, such as polarity, molecular weight, and log Kow—the logarithm of the octanol–water partition coefficient, which reflects the balance between hydrophilicity and lipophilicity and thus influences how antibiotics interact with the environment and living organisms, as well as environmental conditions and species-specific plant traits including root architecture, exudate composition, and enzymatic repertoire [11,34].

In the context of antibiotic pollution, phytoremediation can be categorized into several principal pathways.

### 3.1. Phytoextraction: Uptake and Accumulation

Phytoextraction entails the uptake of antibiotic molecules by plant roots, followed by their accumulation in root tissues or translocation to aerial parts. The mechanisms involved—ranging from passive diffusion to facilitated transport and active uptake—are governed by physicochemical properties of the compound, including lipophilicity, charge state, and molecular size [35,36,37]. Once internalized, antibiotics may be compartmentalized within vacuoles to mitigate toxicity, bound to cell wall components or intracellular proteins, or subjected to partial enzymatic modification prior to sequestration. Certain plant species exhibit remarkable accumulation capacities: *Lemna minor* L. has been shown to efficiently sequester sulfonamides, while *Zea mays* L. demonstrates high uptake of quinolones. However, the efficiency of translocation from roots to shoots depends largely on the mobility of the compound and the plant’s transpiration rate [38,39].

### 3.2. Phytodegradation: Metabolic Breakdown

Phytodegradation denotes the enzymatic transformation of antibiotics into less harmful or more polar metabolites within plant tissues [40,41]. This detoxification process is primarily mediated by oxidative enzymes—including peroxidases, laccases, and cytochrome P450s—that facilitate reactions such as hydroxylation and demethylation. In addition, conjugative enzymes such as glutathione S-transferases catalyze the conjugation of antibiotics with glutathione or sugars, thereby promoting detoxification and storage [42,43,44,45]. Although phytodegradation can markedly decrease the bioavailability of antibiotics, the metabolic fate and ecotoxicological implications of the resulting transformation products remain insufficiently understood. Evidence indicates that phase I and II metabolism in plants resembles hepatic biotransformation in animals, albeit with lower substrate specificity [46].

### 3.3. Rhizodegradation: Microbial Degradation in the Rhizosphere

The rhizosphere—the narrow zone of soil influenced by root activity—plays a central role in antibiotic degradation. Plants release an array of exudates, including sugars, amino acids, and phenolic compounds, which stimulate the proliferation and activity of rhizosphere microorganisms capable of biodegrading antibiotics [47]. In soil-based and wetland systems, rhizodegradation frequently emerges as the dominant pathway of attenuation. Plant species such as *Phragmites australis* (Cav.) Trin. ex Steud. and *Typha latifolia* L. host diverse microbial consortia that can effectively degrade sulfonamides, tetracyclines, and fluoroquinolones [48,49,50]. The synergistic interplay between plant-derived exudates and microbial catabolic genes not only enhances antibiotic removal but also helps mitigate selective pressures that drive antimicrobial resistance [51].

### 3.4. Phytostabilization and Immobilization

Phytostabilization entails the retention of antibiotics within the root zone through mechanisms such as sorption onto root surfaces, binding to soil organic matter, or precipitation. By reducing antibiotic mobility and bioavailability, this process limits their dispersal into non-target environments and decreases the risk of uptake by food crops [21,52]. Although not a degradative pathway, phytostabilization is particularly valuable for compounds with low biodegradability or in contexts where containment is preferable to transformation. Root exudates may further enhance immobilization by altering soil pH or ionic strength, thereby increasing sorption efficiency [53,54].

### 3.5. Phytovolatilization: Rare in Antibiotic Remediation

Phytovolatilization describes the uptake of contaminants by plants and their subsequent release into the atmosphere through transpiration. While this pathway is well established for volatile organic compounds and metalloids, its contribution to antibiotic remediation is minimal, largely because most antibiotics possess low vapor pressures, high polarity, and ionic character [55,56]. Under certain conditions, however, partially transformed metabolites—particularly those derived from sulfonamides or fluoroquinolones—may exhibit increased volatility and be released through stomata. Likewise, microbial transformations within the rhizosphere can generate semi-volatile intermediates that are subsequently absorbed and volatilized by plants [57,58,59]. Despite these possibilities, current evidence suggests that volatilization fluxes remain negligible and do not substantially contribute to the overall dissipation of antibiotics.

### 3.6. Conclusions

In summary, phytoremediation operates through a spectrum of complementary mechanisms—ranging from direct uptake and metabolic transformation within plant tissues to microbially driven degradation and stabilization in the rhizosphere. While each pathway presents unique advantages and constraints, their combined action underpins the capacity of plants to mitigate antibiotic pollution. Having outlined these core mechanisms, the following sections shift focus to plant species—both aquatic and terrestrial—that exemplify the practical application of these processes in real-world contexts.

## 4. Plant Players: Species with High Remediation Potential

Selecting appropriate plant species is fundamental to the success of phytoremediation. For instance, *Brassica juncea* removes tetracycline and oxytetracycline from soil, while *Helianthus annuus* accumulates sulfonamides and fluoroquinolones [10].

Optimal candidates exhibit vigorous growth, high biomass production, tolerance to antibiotic stress, and the capacity for efficient contaminant uptake and transformation. The effectiveness of phytoremediation is often species-specific and strongly influenced by the physicochemical characteristics of the target antibiotic, including polarity, solubility, and persistence [55].

### 4.1. Aquatic Macrophytes

Aquatic plants are particularly well suited for the remediation of contaminated surface waters, municipal and industrial wastewater, and effluents from aquaculture or pharmaceutical production. Their continuous contact with water, rapid growth rates, and capacity to support diverse microbial communities make them especially effective in constructed wetland systems [60,61,62]. *Lemna minor* L., a small floating species with rapid vegetative reproduction, efficiently accumulates antibiotics such as sulfamethoxazole and tetracycline owing to its high surface area-to-volume ratio. *Eichhornia crassipes* (Mart.) Solms., commonly known as water hyacinth, combines prolific biomass production with strong pollutant uptake capacity and has been shown to remove ciprofloxacin, erythromycin, and amoxicillin while simultaneously supporting rhizospheric microbial consortia. *Phragmites australis* (Cav.) Trin. ex Steud., the common reed, is one of the most widely applied species in constructed wetlands; its extensive root system enhances oxygenation and stimulates microbial activity, thereby promoting the degradation of sulfonamides and fluoroquinolones through rhizodegradation [19,63].

### 4.2. Terrestrial Plants

Having discussed the role of aquatic macrophytes in antibiotic remediation, we now turn to terrestrial plant species. These taxa are particularly relevant for the remediation of soils and agricultural environments, where antibiotics often accumulate due to manure application, irrigation with contaminated water, or deposition from effluents. Terrestrial plants can be integrated into agricultural systems, buffer strips, and green infrastructures, providing an important complement to aquatic species. Terrestrial species are equally important for soil remediation and can be integrated into agricultural systems, buffer strips, or green infrastructure [64]. *Brassica juncea* (L.) Czern., or Indian mustard, combines high biomass production with an extensive root system and is capable of uptaking tetracycline and oxytetracycline from soil while tolerating moderate antibiotic concentrations [65,66,67,68,69]. The sunflower (*Helianthus annuus* L. subsp. *annuus*), with its large root network, is particularly suited for phytoextraction; it can accumulate and potentially transform a range of antibiotics while simultaneously producing valuable harvestable biomass [70,71]. *Zea mays* L., or maize, is also relevant in agricultural phytoremediation studies, demonstrating significant antibiotic uptake when irrigated with contaminated water and offering potential applications in managed crop rotation systems for soil detoxification [72,73].

When selecting species for antibiotic phytoremediation, several criteria are critical: tolerance to antibiotic stress, root surface area and exudation capacity, biomass yield and growth rate, adaptability to environmental conditions such as pH, salinity, and climate, and the ability to associate with effective rhizosphere microbiota [74]. Polyculture systems that integrate multiple plant species can further enhance remediation performance by diversifying uptake mechanisms and microbial associations, thereby increasing overall system resilience and efficiency [75].

Together, aquatic and terrestrial species illustrate the versatility of phytoremediation strategies across environmental matrices. Whereas aquatic macrophytes dominate in water-based systems and constructed wetlands, terrestrial plants extend the benefits of phytoremediation to soils and croplands. Both groups demonstrate species-specific adaptations that determine antibiotic uptake, transformation, and microbe-assisted degradation, highlighting the importance of plant selection for effective remediation species.

### 4.3. Criteria for Plant Selection

When selecting plant species for the phytoremediation of antibiotics, several critical criteria must be taken into account. Ideal candidates should exhibit tolerance or resistance to antibiotic stress, possess extensive root surface area coupled with strong exudation capacity, and demonstrate high biomass production with rapid growth. Equally important is their adaptability to environmental factors such as soil pH, salinity, and climate, as well as their ability to establish beneficial associations with rhizosphere microbiota [52]. Incorporating multiple species in polyculture systems can further enhance remediation efficiency by diversifying uptake pathways and microbial interactions, thereby improving overall system resilience and stability [76,77,78]. While plant traits are crucial, their remediation capacity is strongly influenced by interactions with the rhizosphere microbiome. The following section explores the pivotal role of root-associated microorganisms in enhancing antibiotic degradation.

## 5. Beyond the Roots: Rhizosphere Dynamics and Microbial Allies

While plants form the cornerstone of phytoremediation, their effectiveness is greatly amplified by the activity of microorganisms inhabiting the rhizosphere. This symbiotic interaction, known as rhizoremediation, represents one of the most dynamic and promising strategies in sustainable environmental remediation [79]. Rhizosphere-associated microbes not only facilitate the degradation of antibiotics and other micropollutants but also enhance plant health and resilience, thereby strengthening overall system performance [80].

### 5.1. Rhizosphere Microbiota: Catalysts of Bioremediation and Plant Immunity

The rhizosphere, a thin soil zone surrounding plant roots, functions as a dynamic biogeochemical reactor shaped by root exudates such as sugars, amino acids, organic acids, phenolics, and signaling molecules. These compounds fuel microbial growth and activity, fostering a rich microbial community capable of transforming environmental contaminants. In the presence of antibiotics, rhizo-spheric microorganisms can biodegrade these compounds via hydrolysis, oxidation, and demethylation, mineralize them into CO_2_, water, and inorganic ions, or convert them into less toxic intermediates [81,82].

Among the key microbial players are plant growth-promoting microorganisms (PGPMs), which exert both direct and indirect biocontrol effects. Acting as natural pesticides, PGPMs suppress pathogens through mechanisms such as hyper-parasitism, nutrient competition, antimicrobial compound production, and the stimulation of systemic plant defenses [83]. These microbes also induce defensive priming, a physiological state enabling plants to mount faster and stronger responses to subsequent stressors [84]. Representative species include *Pseudomonas fluorescens*, *Bacillus subtilis*, and *Trametes versicolor*, all capable of degrading fluoroquinolones and sulfonamides. Central to this priming are two major immunity pathways. Induced Systemic Resistance (ISR) enhances the plant’s innate immunity without directly attacking pathogens, mediated by microbial signals including lipopolysaccharides (LPS), volatile organic compounds (VOCs), flavonoids, and phytohormones [85,86]. Systemic Acquired Resistance (SAR) is a salicylic acid-dependent response typically activated following infection or microbial interaction. Both ISR and SAR result in the production of defense-related molecules such as reactive oxygen species (ROS), phytoalexins, and pathogenesis-related (PR) proteins, which limit pathogen spread and strengthen plant resistance [87,88,89].

Several studies illustrate these mechanisms. For instance, *Bacillus subtilis* strain BU412 enhances tobacco immunity against *Pseudomonas syringae* by upregulating defense enzymes such as SOD, POD, PPO, and PAL [90]. *Bacillus vallismortis* TU-Oraga21 has been shown to suppress rice blast both directly and through the stimulation of plant defenses [91]. *Pseudomonas fluorescens* SP007s improves resistance in kale and soybean by increasing endogenous salicylic acid levels and degrading quorum-sensing signals from *Xanthomonas axonopodis* via carbamoylphosphate synthetase activity [72,92]. Together, these microbial activities not only accelerate the detoxification of environmental pollutants but also fortify plant defense systems, positioning PGPMs as crucial agents at the intersection of environmental remediation and sustainable crop protection.

### 5.2. Microbial Communities Involved

Antibiotic degradation in soil and aquatic environments is mediated by a wide range of microbial taxa with specialized metabolic capabilities. Prominent bacterial genera such as *Pseudomonas*, *Bacillus*, and *Acinetobacter* are able to degrade sulfonamides, tetracyclines, and β-lactams through enzymatic processes including hydrolysis, oxidation, and demethylation. *Actinomycetes*, a group of filamentous bacteria, also contribute significantly by producing extracellular enzymes that facilitate the breakdown of complex organic compounds.

Fungal communities, particularly ligninolytic fungi, play an equally critical role in the degradation of persistent antibiotics such as fluoroquinolones. Species including *Trametes versicolor* (L.) Lloyd, *Bjerkandera adusta* (Willd.) P. Karst., *Porostereum spadiceum* (Pers.) Hjortstam & Ryvarden, and *Pleurotus ostreatus* (Jacq.) P. Kumm. exhibit strong degradation capacities for ciprofloxacin, norfloxacin, and levofloxacin, mediated by oxidative enzymes such as laccases, heme peroxidases, and cytochrome P450s [93]. In addition, certain fungi can immobilize antibiotics—including norfloxacin and danofloxacin—through biosorption, binding them to cell wall components.

Mycorrhizal fungi, although not always directly involved in antibiotic degradation, influence contaminant fate by altering root exudation patterns and promoting sequestration within the rhizosphere. Through their symbiotic interactions with plant roots, they indirectly shape microbial community composition and enhance the overall detoxification potential [94].

Recent evidence demonstrates that microbial consortia, composed of both bacteria and fungi, often outperform single-species systems. Their superior efficiency arises from metabolic complementarity, shared enzymatic pathways, and horizontal gene transfer, which together enable consortia to degrade complex antibiotic structures and adapt to resistance traits [95]. For example, fungal–bacterial consortia have been shown to completely remove multiple fluoroquinolones within just 15 days under optimized conditions.

Horizontal gene transfer (HGT) within microbial communities, particularly in wastewater environments, further accelerates the dissemination of antibiotic-degrading genes alongside resistance determinants. This highlights the ecological importance of biofilm-forming microbial networks, where close physical proximity promotes both enzymatic cooperation and genetic exchange [96].

In summary, antibiotic degradation in the environment is not the product of isolated microbial activity but rather the outcome of complex and dynamic microbial networks. Bacteria, fungi, and plant-associated symbionts interact synergistically to transform and attenuate pollutants, underscoring the central role of community-level interactions in sustainable remediation processes.

### 5.3. Mechanisms of Plant–Microbe Synergy

The synergistic interactions between plant roots and rhizospheric microorganisms are central to the effectiveness of phytoremediation, particularly in the degradation and transformation of antibiotics and other micropollutants. This synergy operates through multiple interconnected mechanisms. Root exudates—comprising sugars, amino acids, phenolics, and organic acids—can modify the physicochemical properties of the surrounding soil, altering pH and redox conditions in ways that increase the solubility and mobility of antibiotics, thereby enhancing their bioavailability for microbial degradation. Soil pH and redox conditions strongly affect degradation, with acidic pH enhancing antibiotic solubility and redox gradients promoting both aerobic and anaerobic pathways [97]. Certain exudate-derived compounds, such as flavonoids and phenolic acids, also act as biochemical inducers of microbial catabolic pathways, stimulating the expression of degradative enzymes including monooxygenases, laccases, and β-lactamases. Transcriptomic studies confirm that the presence of these compounds can upregulate genes associated with antibiotic degradation, underscoring the direct signaling relationships between plants and microbes [98].

Root architecture further provides a structured, nutrient-rich habitat that promotes microbial colonization and biofilm formation. Microbial biofilms on root surfaces protect communities from environmental stressors, while simultaneously enhancing enzymatic activity and metabolic stability [3]. Plants also exert a strong selective pressure on their microbial partners through the so-called “biased rhizosphere” effect, whereby targeted exudation patterns preferentially enrich microbial taxa capable of antibiotic degradation and detoxification [15]. Within this niche, microbial strains with advantageous fitness traits—including antibiotic resistance, oxidative stress tolerance, and strong colonization capacity—are favored, while the high density of microbial interactions facilitates horizontal gene transfer. Such gene exchange accelerates the dissemination of catabolic functions and resistance mechanisms, further strengthening the degradative potential of the rhizosphere [15].

In addition, plant roots support complex microbial consortia in which metabolic complementarity enhances pollutant breakdown. Ligninolytic fungi such as *Trametes versicolor* and *Pleurotus ostreatus* initiate the degradation of fluoroquinolones through oxidative enzymes, while co-inhabiting bacteria mineralize the resulting intermediates, achieving more complete detoxification [93].

Through these processes, the rhizosphere functions as a dynamic biochemical reactor where coordinated plant and microbial activities drive the transformation and attenuation of environmental contaminants. Advancing our understanding of plant–microbe signaling and metabolic cross-talk is therefore essential for optimizing phytoremediation strategies and for designing synthetic consortia with enhanced catabolic capacity.

### 5.4. Engineering the Rhizosphere

Recent advances in microbial ecology and synthetic biology have paved the way for engineered rhizospheres, where selected or genetically modified microorganisms are introduced to enhance the degradation of antibiotics and related pollutants. Several complementary strategies have been explored.

Bioaugmentation focuses on the deliberate introduction of efficient antibiotic-degrading strains into the rhizosphere. A notable example is *Pseudomonas fluorescens* HK44, a genetically engineered strain capable of degrading polycyclic aromatic hydrocarbons (PAHs) while simultaneously emitting bioluminescent signals in response to contaminant bioavailability. Such dual-function biosensors provide valuable tools for real-time monitoring of remediation processes.

Biostimulation, by contrast, seeks to enhance the activity of indigenous microbial communities through the addition of nutrients or co-substrates, such as organic acids and sugars, which act as metabolic inducers. For instance, inoculation of the halophyte *Salicornia europaea* L. var. *fruticosa* L. with *Brevibacterium casei* and *Pseudomonas oryzihabitans* has been shown to reprogram plant metabolism under stress conditions, leading to increased production of stress-alleviating compounds including caffeic acid, quercetin, and sucrose in both controlled and field environments.

Microbiome engineering represents a further step, aiming to shape rhizospheric microbial communities through selective pressures, co-cultivation strategies, or the manipulation of root exudates. Plant growth-promoting bacteria (PGPB) can modulate the plant metabolome, enhancing the synthesis of antioxidants, chelators, and detoxifying secondary metabolites, thereby improving both the bioavailability and the transformation of antibiotic pollutants [99].

In addition, the application of genetically engineered microorganisms (GEMs) has opened new avenues for targeted bioremediation. These organisms can be designed to express specific catabolic pathways or biosensors that detect, transform, or immobilize pollutants. For example, insertion of the merA gene into *Escherichia coli* enables the reduction in toxic mercury, while recombinant strains of *Deinococcus radiodurans* and *Pseudomonas putida* have been successfully employed for the degradation of organochlorine pesticides [100].

Together, these approaches allow the creation of customized rhizosphere ecosystems capable of sustaining robust and reproducible antibiotic degradation, even under high contaminant loads or in the presence of complex mixtures. The synergistic interplay between stress-tolerant plants and engineered microbial consortia represents a promising frontier at the intersection of environmental remediation and sustainable agriculture.

### 5.5. Challenges and Future Needs

Despite the considerable promise of rhizoremediation as a sustainable approach for mitigating antibiotic contamination, several critical challenges continue to hinder its large-scale implementation and long-term effectiveness. One of the foremost obstacles is the limited persistence of introduced microbial strains. In dynamic and heterogeneous soil ecosystems, engineered or augmented microbial consortia often struggle to compete with native microbiota, restricting their establishment, functional stability, and overall remediation potential under field conditions [25].

Equally complex are the interactions among plants, rhizospheric microorganisms, and antibiotics, particularly when transformation products or co-contaminants such as heavy metals are present. These multifaceted relationships complicate the development of predictable and reproducible remediation strategies and are further complicated by the potential for antibiotic resistance gene (ARG) transfer within microbial communities [101]. A further challenge lies in the poorly understood ecotoxicological profiles of pharmaceutical transformation products (TPs). Generated through microbial metabolism, hydrolysis, or photolysis, many TPs remain insufficiently characterized in terms of environmental fate and toxicity. Alarmingly, some display equal or even greater toxicity than their parent compounds, posing additional risks to both microbial functionality and plant health [102]. Selective pressures also present a significant barrier. Sub-lethal concentrations of antibiotics and their transformation products in the environment act as drivers of ARG proliferation, threatening the stability of remediation-associated microbial communities while raising broader public health concerns related to antimicrobial resistance (AMR) [103].

Addressing these challenges requires a systems-level research perspective. Advances in multi-omics technologies—including metagenomics, transcriptomics, proteomics, and metabolomics—are providing transformative insights into the functional dynamics of plant–microbe–pollutant interactions [104,105]. These integrative approaches facilitate the identification of keystone microbial taxa, regulatory networks, and metabolic pathways that underpin contaminant degradation and ecosystem resilience.

Equally promising is the development of synthetic microbial communities (SynComs) tailored to specific plant genotypes and soil conditions. When combined with knowledge of root exudate–mediated microbiome recruitment, such designed consortia could deliver robust, targeted, and field-adaptable solutions for contaminant removal [106].

In conclusion, overcoming the limitations of rhizoremediation will require cross-disciplinary collaboration, long-term field validation, and regulatory frameworks that safeguard ecological integrity while ensuring remediation efficacy. Integrating strategies to mitigate AMR is equally critical, as protecting environmental and public health must remain a central objective in the advancement of sustainable bioremediation technologies.

### 5.6. Conclusions

The rhizosphere functions as a dynamic interface where plants and microorganisms cooperate to degrade, immobilize, or transform antibiotics. Bacteria, fungi, and engineered microbial consortia significantly enhance plant-based removal, while plants, in turn, modulate microbial activity through root exudates and habitat provision. This synergy elevates phytoremediation from a plant-centric process to a truly integrated biological system. Building on these interactions, the next section explores how nature-inspired engineering solutions such as constructed wetlands translate these principles into scalable applications. By elucidating the synergy between plants and microorganisms, rhizoremediation emerges as a dynamic system. Building on these insights, the next section examines how these principles are translated into scalable applications through constructed wetlands and integrated systems.

## 6. Nature-Inspired Engineering: Constructed Wetlands and Integrated Systems

Laboratory research has clarified plant-based removal mechanisms, but applying them in practice requires careful design, scalability, and system integration. Constructed wetlands are a leading phytoremediation technology, providing a sustainable, low-cost, and versatile solution for treating antibiotic-contaminated water [107].

### 6.1. Constructed Wetlands: Types and Principles

Constructed wetlands are engineered ecosystems designed to replicate the structure and function of natural wetlands, integrating substrate, vegetation, and microbial communities for the treatment of wastewater [108]. They are generally classified into two main types. Free Water Surface (FWS) wetlands are characterized by water flowing across a vegetated substrate, where emergent species such as *Phragmites australis* and *Typha latifolia* promote contaminant uptake and facilitate sedimentation [46]. In contrast, Subsurface Flow (SSF) wetlands direct water through a porous medium such as gravel or sand, with vertical or horizontal flow patterns that enhance interaction with plant roots and rhizospheric microbial communities [109]. Both systems achieve pollutant reduction through a combination of rhizodegradation, microbial metabolism, filtration, sedimentation, and adsorption processes [110].

### 6.2. Performance for Antibiotic Removal

Multiple studies have demonstrated the effectiveness of constructed wetlands (CWs) in removing antibiotics from a variety of effluent sources, including municipal wastewater, aquaculture discharge, and agricultural runoff. Reported removal efficiencies range from 30% to over 90%, depending on design parameters, hydraulic retention time, redox dynamics, and plant species [110,111]. Among the most extensively studied antibiotic classes in these systems are sulfonamides, tetracyclines, and fluoroquinolones. Vertical subsurface flow (VSSF) wetlands, as well as hybrid configurations that combine VSSF with free water surface (FWS) systems, often achieve superior removal due to improved aeration and greater interaction between pollutants and rhizospheric microbial communities [112]. Moreover, the establishment of redox gradients within the rhizosphere—ranging from oxic zones near root surfaces to anoxic zones deeper in the substrate—facilitates both aerobic and anaerobic degradation pathways, thereby enhancing overall contaminant removal [113].

### 6.3. Design Factors and Optimization Strategies

The performance of constructed wetlands can be significantly enhanced through the careful optimization of both system design and biological components. Extending hydraulic retention time generally improves pollutant removal, although this must be balanced against overall treatment capacity. Substrate composition is also critical: materials such as biochar, zeolite, and fine gravel not only enhance adsorption but also provide favorable niches for microbial colonization. Vegetation choice further shapes system efficiency, with species such as *Juncus effusus* L. subsp. *effusus*, *Phragmites australis*, and even highly effective invasive macrophytes like *Eichhornia crassipes* contributing extensive root systems that foster microbial diversity and facilitate pollutant transformation. Flow configuration exerts an additional influence, with hybrid systems that combine vertical and horizontal flows achieving superior treatment outcomes, particularly for complex mixtures of antibiotics and other pharmaceuticals [114].

### 6.4. Integration with Other Technologies

To further enhance pollutant removal and broaden system functionality, constructed wetlands can be integrated with complementary green technologies. The addition of biochar amendments improves antibiotic adsorption while simultaneously supporting microbial metabolic activity [115]. Photocatalytic treatments, such as solar-driven TiO_2_, can be applied upstream to degrade recalcitrant compounds before water enters the wetland [116]. Floating treatment wetlands (FTWs), using plants such as *Lemna minor* or *Eichhornia crassipes*, are particularly effective in decentralized systems and stagnant water bodies [117]. Coupling constructed wetlands with microbial fuel cells provides the dual benefit of enhanced pollutant oxidation and energy recovery [118]. Collectively, these hybrid strategies combine ecological processes with engineered precision, offering powerful tools to address emerging contaminants more efficiently.

### 6.5. Case Studies and Real-World Applications

Constructed wetlands have been successfully deployed across diverse geographical regions and use scenarios (Table 2). In China, a hybrid CW designed for swine farm effluent achieved over 80% removal of tetracyclines using *Phragmites* and *Typha* established on gravel–sand beds [119]. In Europe, floating treatment wetlands (FTWs) have demonstrated substantial reductions in oxytetracycline and enrofloxacin from aquaculture effluents. In Sub-Saharan Africa, low-cost vertical flow wetlands are being applied to treat healthcare wastewater containing sulfonamides and β-lactams, achieving promising removal efficiencies. These examples underscore the adaptability and resilience of constructed wetlands under widely varying environmental and socioeconomic contexts.

As nature-based ecological engineering solutions, CWs offer cost-effective and decentralized strategies for mitigating antibiotic pollution. However, challenges remain, including long-term substrate saturation, seasonal fluctuations in performance, and the safe disposal of contaminated biomass. Addressing these limitations will be essential for advancing constructed wetlands from localized demonstration projects to globally recognized best practices.

**Table 2 antibiotics-14-01031-t002:** Case studies of constructed wetlands applied for the removal of antibiotic-contaminated water in different regions. The table summarizes target antibiotics, main findings on treatment performance, and corresponding references, highlighting the versatility of constructed wetlands as nature-based solutions across diverse environmental contexts.

Region	Target Antibiotic(s)	Main Findings/Application Details	Reference(s)
China	Tetracyclines (TC, OTC, CTC)	Vertical up-flow CWs treating swine wastewater achieved high removal (69–99.9%) of tetracyclines and reduction in tet genes.	Huang et al. [120]
China	Oxytetracycline, Tetracycline, Doxycycline, Chlortetracycline	CWs filled with coke + plants (CP-CW) showed removal rates of ~91% for OTC, ~90% for TC, ~85% for DOX.	Bai et al. [13]
Europe (Poland & Czechia)	Sulfonamides (e.g., sulfamethoxazole)	In full-scale CWs, sulfamethoxazole was removed with ~86–99% efficiency; sul1 genes persisted with little change.	Felis et al. [121]
Lab-scale study	Oxytetracycline	Vertical-flow CWs with zeolite + activated carbon achieved up to 97% removal of OTC.	Yuan et al. [122]

## 7. Challenges and Knowledge Gaps

Despite considerable progress in phytoremediation and plant–microbe-based approaches for antibiotic removal, several scientific, technical, and regulatory challenges remain unresolved (Table 3). Addressing these knowledge gaps is essential to advance from proof-of-concept experiments to robust, field-deployable systems.

### 7.1. Incomplete or Variable Removal Efficiency

Phytoremediation performance is inherently variable, influenced by antibiotic structure and concentration, plant species, environmental conditions, and the composition of microbial communities. Polar and persistent compounds such as fluoroquinolones and macrolides are particularly difficult to remove because of low bioavailability, strong sorption to soil particles, or microbial recalcitrance [61]. Environmental fluctuations in pH, redox potential, temperature, and organic matter further affect microbial activity and enzymatic degradation rates [123]. Consequently, highly polar or strongly adsorbed compounds may resist both plant uptake and microbial transformation, resulting in partial or negligible removal.

### 7.2. Phytotoxicity and Growth Inhibition

High antibiotic concentrations can impair plant physiology, reducing biomass, altering morphology, and suppressing photosynthetic activity. For example, tetracyclines and sulfonamides disrupt chloroplast function, interfere with nitrogen metabolism, and alter elemental stoichiometry in plant tissues [124]. Such phytotoxic effects present significant barriers to long-term phytoremediation, particularly in high-load environments such as livestock effluents or heavily contaminated soils.

### 7.3. Accumulation in Plant Tissues and Biomass Management

When uptake exceeds degradation, antibiotics can accumulate in roots or aerial tissues, raising concerns about secondary contamination through biomass reuse, leaching during rainfall, or trophic transfer via herbivores [125]. The absence of standardized guidelines for biomass disposal, especially in low- and middle-income countries, further amplifies ecological and human health risks [8].

### 7.4. Unknown Transformation Products and Ecotoxicity

Although microbial and enzymatic pathways can degrade antibiotics, the resulting transformation products (TPs) are often poorly characterized. Some exhibit greater toxicity, mobility, or persistence than their parent compounds, with the potential to exacerbate the spread of antibiotic resistance [126]. Analytical limitations hinder routine detection, and the scarcity of ecotoxicological data for these metabolites prevents comprehensive risk assessment [127].

### 7.5. Complexity of Mixed Contaminant Environments

Antibiotics frequently co-occur with pharmaceuticals, pesticides, heavy metals, and organic matter. These mixtures can generate synergistic or antagonistic effects that complicate predictions of degradation pathways and treatment performance [128]. For example, heavy metals may inhibit key microbial degraders, whereas organic co-substrates can shift microbial community dynamics or promote co-metabolic degradation [129].

### 7.6. Limited Field-Scale Validation and Long-Term Studies

Most current knowledge derives from greenhouse or mesocosm experiments [130]. Field-scale studies remain scarce and often lack multi-season monitoring, assessments of microbial resilience, or economic feasibility analyses. Moreover, the long-term effects of antibiotic exposure on soil fertility, microbial evolution, and plant fitness are still poorly understood [131].

### 7.7. Regulatory and Standardization Gaps

The application of phytoremediation for antibiotic removal is constrained by the absence of dedicated regulatory standards and monitoring frameworks [52]. Key shortcomings include the lack of unified environmental thresholds for antibiotic residues in soil and water, standardized validation protocols for phytoremediation, and tools for evaluating risks associated with transformation products, antibiotic resistance genes (ARGs), and leachates [132]. Notably, many antibiotics used in agriculture persist in the environment and interfere with essential endosymbiotic functions in plants and animals, including chloroplast and mitochondrial activity. These findings highlight the urgent need for stringent, globally coordinated regulatory measures [133].

**Table 3 antibiotics-14-01031-t003:** Summary of phytoremediation strategies, detailing the targeted issues, proposed solutions, underlying mechanisms, representative plant species, roles of the rhizosphere, engineered systems, advantages, limitations, and future perspectives. The second column presents concise summaries, while the third column lists the supporting references.

Aspect	Summary	References
Addressed Problem	Environmental dissemination of antibiotics → ecological imbalances, spread of resistance genes (ARGs), inefficiency/high cost of conventional treatment technologies.	Bielen et al. [1]; Xu et al. [2]; Gomes [3]; Grenni et al. [4]; Tang et al. [5]; La Rosa et al. [6]; Huang et al. [7].
Proposed Solution	Phytoremediation: use of plants (and associated microbes) to remove, transform, or immobilize antibiotics in soils and waters.	Singh et al. [8]; Aryal [9]; Kafle et al. [10].
Main Mechanisms	- Phytoextraction (uptake and accumulation); - Phytodegradation (plant enzymatic transformation); - Rhizodegradation (microbial degradation in the rhizosphere); - Phytostabilization (immobilization in rhizoplane/rhizosphere).	Chen et al. [19]; Zhao et al. [20]; Zhou et al. [21]; Yang et al. [22]; Li et al. [23]; Yi et al. [24].
Key Plant Species	- Aquatic: *Lemna minor*, *Eichhornia crassipes*, *Phragmites australis*; - Terrestrial: *Brassica juncea*, *Zea mays*, *Helianthus annuus*.	Alfarsi et al. [62]; Von Salzen et al. [63]; Aydin et al. [64]; Xu et al. [66]; Huang et al. [67]; Borowik et al. [72]; Madikizela [77]; Milke et al. [78].
Role of the Rhizosphere	Root-associated microbes (bacteria, fungi, actinomycetes) enhance degradation; plant–microbe synergies are crucial for efficiency.	Kraemer et al. [48]; Martínez-Martínez et al. [49]; Zambelli et al. [50]; Angelini [51]; Akrout et al. [94]; Mohanram & Kumar [98].
Engineered Systems	Constructed wetlands (CWs), floating treatment wetlands (FTWs), integration with biochar, photocatalysis, microbial fuel cells.	He et al. [12]; Bai et al. [13]; Sabri et al. [110]; Tarigan et al. [111]; Ajibade et al. [115]; Chen et al. [116]; Alcaide et al. [118].
Advantages	Nature-based, low-cost, eco-friendly solution; co-benefits: increased biodiversity, soil stabilization, improved water quality.	Haque et al. [18]; Chowdhury et al. [31].
Limitations	- Variable efficiency; - Phytotoxicity at high concentrations; - Antibiotic accumulation in plant tissues; - Poorly known and potentially toxic transformation products; - Few field-scale and long-term validations.	Narciso et al., 2023 [124]; Minden et al., 2018 [125]; Polianciuc et al., 2020 [126]; Zhang et al., 2025 [127]; Moreno et al., 2022 [131]
Future Perspectives	- Genetic engineering of plants and microbes; - Rhizosphere microbiome optimization; - Smart monitoring (biosensors, AI); - Hybrid and modular systems; - Integration into circular economy and environmental regulations.	Mallari et al., 2025 [134]; Diwan et al., 2022 [135]; Janga et al., 2023 [136]; Agrahari et al., 2024 [137]; Zaman et al., 2024 [138]; Longo et al., 2024 [139].

## 8. The Road Ahead: Innovation and Future Research Directions

To fully harness the potential of plant-based technologies for mitigating antibiotic pollution, future research must move beyond descriptive case studies and address the mechanistic, technological, and regulatory gaps that currently limit large-scale deployment [140]. The next generation of phytoremediation systems is expected to be hybrid, data-driven, and specifically tailored to the unique characteristics of environmental matrices and contaminant profiles.

### 8.1. Genetic Engineering and Synthetic Biology

Genetic engineering and synthetic biology hold considerable promise for enhancing the phytoremediation of antibiotics. Potential strategies include overexpressing key degradative enzymes such as cytochrome P450s, peroxidases, and laccases to improve transformation rates [134,141]; modifying root architecture or exudation profiles to stimulate beneficial rhizosphere microbial communities [142]; and designing synthetic biology circuits that activate degradative pathways only in the presence of target antibiotics [143]. While these innovations could significantly improve system efficiency, they must be carefully evaluated for ecological safety, risks of horizontal gene transfer, and societal acceptance.

### 8.2. Microbiome Engineering and Rhizosphere Optimization

Recent advances in metagenomics, metatranscriptomics, and metabolomics now allow for the fine-tuning of plant–microbe interactions [144]. Promising approaches include the design of custom microbial consortia with targeted degradative functions [135], inoculation with locally adapted bacterial strains capable of antibiotic breakdown [9], and selective breeding or engineering of plants that preferentially recruit efficient microbial degraders [145]. These strategies enhance system resilience and accelerate contaminant removal, even in complex and variable environmental matrices.

### 8.3. Smart Monitoring and AI-Driven System Design

The integration of advanced monitoring technologies will be critical to the optimization of future phytoremediation systems. Biosensors and portable mass spectrometry enable continuous tracking of antibiotic concentrations and metabolites, providing early detection of system overload or failure. These data streams can be combined with machine learning and artificial intelligence tools to model performance under diverse conditions, optimize system parameters such as retention time and flow rates, and guide the design of adaptive, site-specific treatment systems [146].

### 8.4. Hybrid and Modular Remediation Systems

Phytoremediation technologies will increasingly be integrated with complementary treatment approaches to improve overall performance [136]. For example, coupling wetlands with biochar or nanomaterials can enhance adsorption and surface-mediated degradation [137], while combining plant systems with photocatalysis (e.g., TiO_2_-based processes) allows oxidative pretreatment before plant-mediated polishing [147]. Modular floating wetlands and containerized phytoreactors offer portable, scalable, and flexible solutions adaptable to urban, agricultural, and remote environments [148].

### 8.5. Policy, Education, and Circular Economy Integration

The advancement of phytoremediation must be supported by clear regulatory frameworks for system validation, monitoring, and approval [149]. Equally important are educational initiatives to train practitioners and raise public awareness [138]. Integration into circular economy models will be essential, ensuring that harvested biomass is safely valorized—for example, through conversion into bioenergy—while minimizing secondary contamination risks [150]. Ultimately, phytoremediation should be viewed not as a stand-alone solution but as part of a multi-barrier, sustainable strategy for antibiotic management and pollution prevention [139,151,152].

## 9. Future Directions and Conclusions: Rooting for Green Remediation

Antibiotic pollution has emerged as a critical global issue, intricately linked to antimicrobial resistance, ecosystem disruption, and declining water quality [52,85]. Plant-based strategies provide a promising, nature-inspired response, offering decentralized, low-energy, and adaptable solutions. Plants contribute to antibiotic removal through direct uptake, metabolic degradation, and stimulation of rhizosphere microbial activity [106,153]. Both aquatic and terrestrial species have been shown to target diverse antibiotic classes [90], while rhizosphere microbiota remain essential for enhancing biodegradation [23]. Constructed wetlands and hybrid systems illustrate scalable potential [154], though challenges such as variable efficiency, phytotoxicity, uncertain transformation products, and biomass management persist [155,156].

Future progress requires shifting from descriptive studies to mechanistically informed, field-validated solutions [140]. Hybrid systems integrating plants with biochar, photocatalysis, or floating wetlands can improve adaptability [136,137,147,148]. Advances in microbiome engineering—through tailored consortia, targeted inoculations, or plants bred to recruit efficient degraders—will strengthen resilience [135,144,145]. Genetically engineered plants may further enhance antibiotic degradation via enzyme overexpression, root exudate modulation, or synthetic biology circuits [134,141,142,143], though ecological safety and societal acceptance must be critically assessed.

Digital technologies will also play a pivotal role: real-time biosensors and portable mass spectrometry enable continuous monitoring, while machine learning can model and optimize system performance under varying conditions [146].

Yet, technological innovation alone is insufficient. Supportive policy frameworks are needed to establish thresholds for antibiotic residues, standardized validation protocols, and robust risk assessment tools [132,149]. Education, capacity building, and integration into circular economy models—where biomass is valorized for energy or resource recovery—will further enhance sustainability [138,150].

Taken together, these developments suggest a transition from phytoremediation as a fragmented research field to a mature component of green infrastructure. Plant-based systems should be understood not as stand-alone solutions but as part of a multi-barrier strategy combining plants, microbes, engineered substrates, and digital monitoring. Ultimately, phytoremediation’s promise lies in uniting ecological restoration with technological innovation. With sustained cross-disciplinary research, field validation, and policy support, it may evolve into a globally deployable tool for safeguarding environmental and public health while advancing sustainable, circular pollution management [21,139,151,152,153,154,155,156,157].

In summary, phytoremediation offers a sustainable, low-cost strategy for mitigating antibiotic pollution. Future efforts should focus on plant–microbe engineering, integration with constructed wetlands, and regulatory frameworks to ensure large-scale applicability.

## Figures and Tables

**Figure 1 antibiotics-14-01031-f001:**
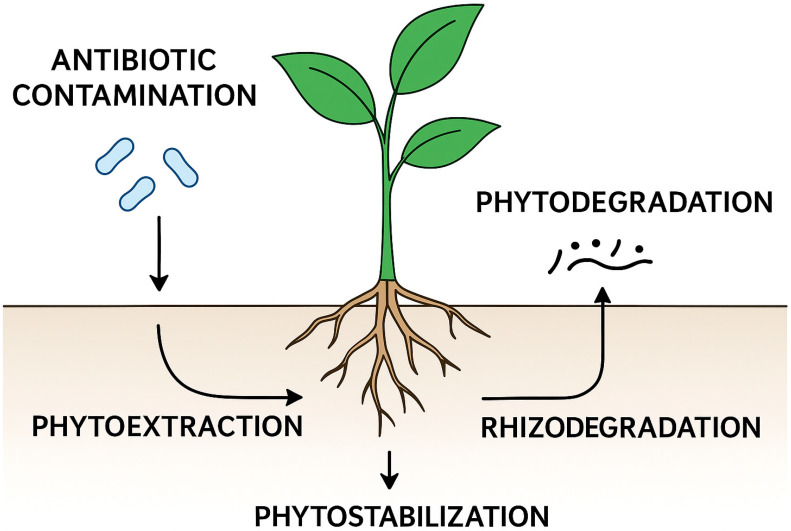
Main phytoremediation pathways involved in the removal of antibiotic contaminants from soil. Antibiotics can be taken up and accumulated in plant tissues (**phytoextraction)**, immobilized in the rhizosphere (**phytostabilization**), degraded within plant tissues through enzymatic processes such as oxidation, reduction and hydrolysis (**phytodegradation**), or broken down by soil microorganisms stimulated by root exudates (**rhizodegradation**).

**Table 1 antibiotics-14-01031-t001:** Inclusion and exclusion criteria for article selection.

Criteria	Inclusion	Exclusion
Language	English	Non-English publications
Type of publication	Peer-reviewed articles (original research, reviews, meta-analyses, case studies)	Conference abstracts, editorials, preprints
Time frame	Published between 2015 and August 2025	Published before 2015
Topic relevance	Focus on plant-based removal or degradation of antibiotics from environment	Articles focused exclusively on metals, pesticides, or non-antibiotic pollutants
Context	Aquatic and terrestrial environmental settings	Purely clinical or pharmacological studies
Mechanisms	Studies exploring plant uptake, degradation, rhizosphere activity, or system design	Studies lacking mechanistic or empirical data

## Data Availability

No new data were created or analyzed in this study. Data sharing is not applicable to this article.

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
