# Peer review of "Green Solutions to a Growing Problem: Harnessing Plants for Antibiotic Removal from the Environment"

_antibiotics, 2025, doi:10.3390/antibiotics14101031_

Round 1

Reviewer 1 Report

Comments and Suggestions for Authors

This review summarizes plant-based solutions to a long-lasting problem, the removal of antimicrobials from the environment. This timely and very important review on phytoremediation approach will therefore attract attention of readers from various fields. Thus, the publication of this very important review is highly recommended. However, there are several minor issues that need to be addressed before the manuscript can be accepted for publication.

  1. Line 200: Please briefly describe “log Kow” for readers from outside of the field.
  2. Lines 288-302 and lines 306-319: These parts are the same. The sentences were copied from above section.
  3. Not all authors are listed in the Authors' contributions section.
  4. There are several typos that must be corrected:

- Abstract, line 18-19: …. has has emerged as an effective and promising …

- Introduction, line 59: …. has has emerged as a sustainable and cost-effective alternative.

- Line 394: Xanthomonas axonopodis

- Line 583: Typha latifolia

Author Response

Comment 1: [Line 200: Please briefly describe “log Kow” for readers from outside of the field].
Response 1: [We thank the reviewer for pointing this out. We agree with the suggestion, and the term log Kow has been described as the logarithm of the octanol–water partition coefficient, which reflects the balance between hydrophilicity and lipophilicity and thus influences how antibiotics interact with the environment and living organisms. This explanation has been included in page 5, paragraph 3.1 Phytoremediation Pathways: How Plants Tackle Antibiotics, lines 200–202].

Comment 2: [Lines 288-302 and lines 306-319: These parts are the same. The sentences were copied from above section].
Response 2: [We thank the reviewer for highlighting this duplication. We agree with the comment, and the manuscript has been revised by removing the repeated text at the beginning of section 4.2 Terrestrial Plants, page 8, lines 308–322].

Comment 3: [Not all authors are listed in the Authors' contributions section].
Response 3: [We are grateful for this observation. The Authors’ Contributions section has been revised accordingly, and the contribution of Gokhan Zengin (G.Z.) has now been included].

Comment 4: [There are several typos that must be corrected].
Response 4: [We thank the reviewer for drawing our attention to these typos. All the indicated errors have been corrected. In the Abstract (page 1, line 18), the repeated word “has” has been removed. In the Introduction (page 2, line 59), the same issue has been corrected by removing the extra “has.” In Section 5.1 Rhizosphere Microbiota: Catalysts of Bioremediation and Plant Immunity (page 10, line 396), the species name Xanthomonas axonopodis has been corrected. Finally, in line 583, the species name Typha latifolia has been properly formatted in italics].

Reviewer 2 Report

Comments and Suggestions for Authors

The authors provide a comprehensive analysis of utilizing key plant species for antibiotic removal from the environment. Further, it discusses the multiple mechanisms- phytoextraction, phytodegradation, rhizodegradation, and phytostabilization, progress and advantages, the challenges, and advanced approaches to tackle existing concerns in phytoremediation.

Plant systems and microbial counterparts as natural agents have been increasingly employed in environmental bioremediation, whether for contaminants/pollutants or removal of toxic substances. The growing incidence of antibiotic resistance in the environment defines a crucial problem and requires immediate attention.

The scope of the topic is timely and promising, suggesting that plants as natural resources are eco-friendly, non-toxic, cheaper, and effective agents for phytoremediation of antibiotics in the environment.

Abstract: It represents the summary of the manuscript. In addition to the examples of plant species, it should prioritize action goals, providing examples of success in phytoremediation, feasible solutions to existing challenges (examples), and future directions in the study.

Apart from the plant-based remediation of antibiotics, what are the other strategies employed for the same? Are these more efficient than plants? Discuss. It is suggested to compare and contrast the emerging approaches employed for the removal of antibiotics from the environment.

Figure 1. The captions discussing the key phytoremediation pathways should be present below the diagram. The mechanisms of antibiotic removal from the soil need to be discussed further- giving examples of soil microbes and mechanisms in rhizodegradation. steps in phytodegradation, examples of antibiotic contamination etc. This will provide a better understanding of the audience.

The article discusses the phytoremediation of antibiotics by plant species. It is important to provide examples of aquatic and terrestrial plant species and the mechanisms of remediation in some details.

In some sections of the manuscript, the information is repeated, please remove the unnecessary duplications. Also, it is important to revise and organize some sections for a systematic and clear presentation. There should be a proper flow of information and the paragraphs should follow a consistent order.

Line 175179: Despite these promising outcomes, plant-based remediation is not universally applicable………………and molecular structure. It is suggested to expand on the limitations of plant-based strategies, discussing key examples of phytoremediation.

An important literature on the same topic discusses the use of advanced synthetic biology approaches in plants to address antimicrobial resistance. It is relevant to the manuscript and may be cited.

Tiwari P, Khare T, Shriram V, Bae H, Kumar V (2021). Plant synthetic biology for producing potent phyto-antimicrobials to combat antimicrobial resistance. Biotechnology Advances, 48, 107729, https://doi.org/10.1016/j.biotechadv.2021.107729.

References are fine, please ensure it follows MDPI guidelines.

Comments on the Quality of English Language

Moderate English revisions are required in the manuscript.

Author Response

Comment 1: [Abstract: It represents the summary of the manuscript. In addition to the examples of plant species, it should prioritize action goals, providing examples of success in phytoremediation, feasible solutions to existing challenges (examples), and future directions in the study].
Response 1: [We thank the reviewer for this constructive suggestion. In the abstract, a portion of the text has been revised to emphasize the high capacity of certain high-performing aquatic and terrestrial species in phytoremediation, while also highlighting goals, challenges, and future directions].

Comment 2: [Apart from the plant-based remediation of antibiotics, what are the other strategies employed for the same? Are these more efficient than plants? Discuss. It is suggested to compare and contrast the emerging approaches employed for the removal of antibiotics from the environment].
Response 2: [We are grateful for this important comment. Apart from plant-based systems, other strategies for antibiotic removal include advanced oxidation processes, membrane filtration, ozonation, UV photolysis, adsorption with biochar or nanomaterials, photocatalysis (e.g., TiOâ‚‚-based), microbial fuel cells, and engineered microbial consortia. These have been discussed in comparison with plant-based systems].

Comment 3a: [Figure 1. The captions discussing the key phytoremediation pathways should be present below the diagram].
Response 3a: [We thank the reviewer for the suggestion. The caption for Figure 1 has been moved directly beneath the figure].

Comment 3b: [The mechanisms of antibiotic removal from the soil need to be discussed further—giving examples of soil microbes and mechanisms in rhizodegradation, steps in phytodegradation, examples of antibiotic contamination etc. This will provide a better understanding of the audience].
Response 3b: [We are grateful for this suggestion. Plants remove antibiotics through four main pathways—uptake, enzymatic degradation, microbe-mediated rhizodegradation, and immobilization. Examples include Phragmites australis, Typha latifolia, Brassica juncea, and Helianthus annuus, supported by key microbes such as Pseudomonas, Bacillus, actinomycetes, and fungi].

Comment 4a: [The article discusses the phytoremediation of antibiotics by plant species. It is important to provide examples of aquatic and terrestrial plant species and the mechanisms of remediation in some details].
Response 4a: [We thank the reviewer for this helpful suggestion. Aquatic species such as Lemna minor, Eichhornia crassipes, and Phragmites australis remove antibiotics via uptake, enzymatic degradation, and rhizosphere-mediated microbial breakdown. Terrestrial plants like Brassica juncea, Helianthus annuus, and Zea mays similarly accumulate and transform tetracyclines and quinolones, supported by root exudates that stimulate microbial consortia].

Comment 4b: [In some sections of the manuscript, the information is repeated, please remove the unnecessary duplications].
Response 4b: [We appreciate this observation and agree with the reviewer. The manuscript has been revised by removing the duplicated text at the beginning of Section 4.2 Terrestrial Plants (page 8, lines 308–322)].

Comment 5: [Also, it is important to revise and organize some sections for a systematic and clear presentation. There should be a proper flow of information and the paragraphs should follow a consistent order].
Response 5: [We thank the reviewer for this valuable advice. To improve the logical flow, we added Section 3.7, a new introduction and final part in Section 4.2, and a concluding part in Section 4.3. Section 5.6 (Conclusions) was also included to provide a systematic closure. In addition, appropriate transitions were introduced to improve readability and consistency between sections].

Comment 6: [Line 175-179: Despite these promising outcomes, plant-based remediation is not universally applicable… and molecular structure. It is suggested to expand on the limitations of plant-based strategies, discussing key examples of phytoremediation].
Response 6: [We are grateful for this comment. The sentence between lines 175–179 has been revised as follows: “Despite these promising outcomes, phytoremediation techniques are often slow and constrained by the depth of contaminants; therefore, they need to be complemented with additional strategies (e.g., microbial augmentation, engineered substrates, hybrid systems) to enhance their effectiveness and scalability”].

Comment 7: [An important literature on the same topic discusses the use of advanced synthetic biology approaches in plants to address antimicrobial resistance. It is relevant to the manuscript and may be cited. Tiwari P, Khare T, Shriram V, Bae H, Kumar V (2021). Plant synthetic biology for producing potent phyto-antimicrobials to combat antimicrobial resistance. Biotechnology Advances, 48, 107729, https://doi.org/10.1016/j.biotechadv.2021.107729].
Response 7: [We thank the reviewer for this relevant suggestion. The reference “Van Der Helm, E.; Genee, H.J.; Sommer, M.O.A. The Evolving Interface Between Synthetic Biology and Functional Metagenomics. Nat. Chem. Biol. 2018, 14(8), 752–759” has been replaced with the recommended citation: “Tiwari P, Khare T, Shriram V, Bae H, Kumar V (2021). Plant synthetic biology for producing potent phyto-antimicrobials to combat antimicrobial resistance. Biotechnology Advances, 48, 107729,
https://doi.org/10.1016/j.biotechadv.2021.107729”].

Comment 8: [References are fine, please ensure it follows MDPI guidelines].
Response 8: [We appreciate this reminder. The references have been carefully revised to ensure that they fully comply with MDPI guidelines].

Reviewer 3 Report

Comments and Suggestions for Authors

The draft showed phytoremediation uses plants and their microbes as a promising nature-based solution to reduce antibiotic pollution and resistance risks. More summaries can be achieved by using figures and tables. The minor revisions are as follows:

Line 18: deleted “has”.

Line 29: It is suggested to incorporate plant-microbe interactions and environmental risks of antibiotic degradation.

Lines 35-37: Are there too many keywords?

Line 88: and “outlook”?

Line 109: The table should be consistent and in three lines.

Lines 131-155: The title of the figure is usually placed at the bottom of the figure, and figures below it are similar.

Line 159 and 167: You mentioned enzymatic reactions. Could you provide some examples?

Line 191: It seems that the subheading "3.1" can be removed as it is not a parallel concept to the subheading below.

Lines 276-286: Some examples of plant species that have such effects should be provided.

Line 378: Some examples of microorganism species that have such effects should be provided.

Line 448: How does pH and redox conditions affect the ability of plants and microorganisms to remove antibiotics?

Line 567: It would be better to summarize this and add a diagram illustrating the mechanism of constructed wetlands?

Line 638: More cases can be provided and summarized in a table.

Line 868: It is best to summarize and write down the conclusion.

Author Response

Comment 1: [Line 18: deleted “has”].
Response 1: [We thank the reviewer for the suggestion; the word
has has been removed from the abstract].

Comment 2: [Line 29: It is suggested to incorporate plant-microbe interactions and environmental risks of antibiotic degradation].
Response 2: [We appreciate the reviewer’s comment; plant-microbe interactions and the environmental risks of antibiotic degradation have been incorporated into the abstract].

Comment 3: [Lines 35-37: Are there too many keywords?].
Response 3: [We thank the reviewer for the observation; the keywords have been reduced to six by removing those from
rizodegradation onwards].

Comment 4: [Line 88: and “outlook”?].
Response 4: [We thank the reviewer for raising this point; after careful checking, we confirm that the word
outlookdoes not appear in any section].

Comment 5: [Line 109: The table should be consistent and in three lines].
Response 5: [We appreciate the reviewer’s suggestion; the table 1 has been reformatted to ensure consistency and presented in three columns. Lines 103-105].

Comment 6: [Lines 131-155: The title of the figure is usually placed at the bottom of the figure, and figures below it are similar].
Response 6: [We thank the reviewer for the helpful advice; the caption of Figure 1 has been moved below the figure, and the same formatting has been applied consistently to the other figures. Lines: 126-150].

Comment 7: [Line 159 and 167: You mentioned enzymatic reactions. Could you provide some examples?].
Response 7: [We appreciate the reviewer’s suggestion; examples of enzymatic reactions have been added to clarify this section. Lines: 154-156].

Comment 8: [Line 191: It seems that the subheading "3.1" can be removed as it is not a parallel concept to the subheading below].
Response 8: [We thank the reviewer for the observation; the subheading
3.1 has been removed and its content integrated into Section 3. Subheading ‘3.1’ is included between lines 184–196.].

Comment 9: [Lines 276-286: Some examples of plant species that have such effects should be provided].
Response 9: [We are grateful for this suggestion; examples of plant species with such effects have been added in Section 4, highlighted in glaucous green: Lines 275-277].

Comment 10: [Line 378: Some examples of microorganism species that have such effects should be provided].
Response 10: [We thank the reviewer for the comment; examples of microorganism species with these effects have been included in the text highlighted in yellow: Lines 372-374].

Comment 11: [Line 448: How does pH and redox conditions affect the ability of plants and microorganisms to remove antibiotics?].
Response 11: [Lines 441-443: We appreciate the reviewer’s question; this section has been revised to clarify how pH and redox conditions influence the ability of plants and microorganisms to remove antibiotics].

Comment 12: [Line 567: It would be better to summarize this and add a diagram illustrating the mechanism of constructed wetlands?].
Response 12: [We thank the reviewer for the constructive suggestion. Lines 563-568: Section 6 has been summarized and a diagram illustrating the mechanism of constructed wetlands has been added].

Comment 13: [Line 638: More cases can be provided and summarized in a table].
Response 13: [We are grateful for the suggestion; additional cases have been provided and summarized in a dedicated Table 2: Lines 649-654].

Comment 14: [Line 868: It is best to summarize and write down the conclusion].
Response 14: [We thank the reviewer for the helpful advice; the conclusions have been summarized accordingly. Lines: 796-832].

Round 2

Reviewer 2 Report

Comments and Suggestions for Authors

Thank you for a careful and thorough revision of the manuscript as per the suggestions.

Congratulations.